# A MIXTURE OF LINEAR CORRECTIONS GENERATES SECURE CODE

## ABSTRACT

Large language models (LLMs) have become proficient at sophisticated code-generation tasks, yet remain ineffective at reliably detecting or avoiding code vulnerabilities. Does this deficiency stem from insufficient learning about code vulnerabilities, or is it merely a result of ineffective prompting? Using representation engineering techniques, we investigate whether LLMs internally encode the concepts necessary to identify code vulnerabilities. We find that current LLMs encode precise internal representations that distinguish vulnerable from secure code–achieving greater accuracy than standard prompting approaches. Leveraging these vulnerability-sensitive representations, we develop an inference-time steering technique that subtly modulates the model's token-generation probabilities through a mixture of corrections (MoC). Our method effectively guides LLMs to produce less vulnerable code without compromising functionality, demonstrating a practical approach to controlled vulnerability management in generated code. Notably, MoC enhances the security ratio of Qwen2.5-Coder-7B by 8.9%, while simultaneously improving functionality on HumanEval pass@1 by 2.1%.

## 1 INTRODUCTION

Large language models (LLMs) have rapidly become useful tools for developers, demonstrating remarkable proficiency across a wide array of code generation tasks Chen et al. (2021); Jiang et al. (2024). Current LLMs excel at understanding complex programming concepts Zheng et al. (2023), generating syntactically correct and functionally relevant code Hui et al. (2024); Zhuo et al. (2025), and even providing explanations, optimizations, and debugging assistance Lewkowycz et al. (2022).

Despite these advances, even state-of-the-art models exhibit significant limitations with identifying vulnerable code. Our empirical analysis (Figure 1) on different sizes of CodeLlama Grattafiori et al. (2023) and Qwen2.5-Coder Hui et al. (2024) reveals that traditional prompting techniques, including few-shot exemplars and detailed Common Weakness Enumeration (CWE) descriptions, result in accuracy comparable to random guessing (50%). Surprisingly, increasing the model parameter count fails to reliably improve detection accuracy, suggesting a persistent gaps between increased coding capabilities and the closely related task of identifying and generating *secure* code.

This motivates the question: *do code-generating LLMs inherently lack the knowledge to differentiate between vulnerable and secure code, or is this knowledge simply not accessible via prompting?* Using linear probing Alain & Bengio (2017); Zou et al. (2023), we find that LLMs do indeed possess latent representations that distinguish secure from vulnerable code far more effectively than standard prompts. Thus, despite these models' apparent lack of proficiency at the task of identifying vulnerable code, it is possible to access models' precise learned knowledge about vulnerabilities to accurately perform identification during inference, without the need for more expensive methods involving fine-tuning Fu & Tantithamthavorn (2022).

Building on this insight, we next investigate whether these latent representations can be leveraged during code generation. Specifically, we explore how to compute *correction vectors*–derived directly from clusters, linear probes, or through auxiliary neural networks–that encode vulnerability distinctions. These vectors, computed separately for individual CWEs, create a *mixture* of precise linear corrections for code generation.

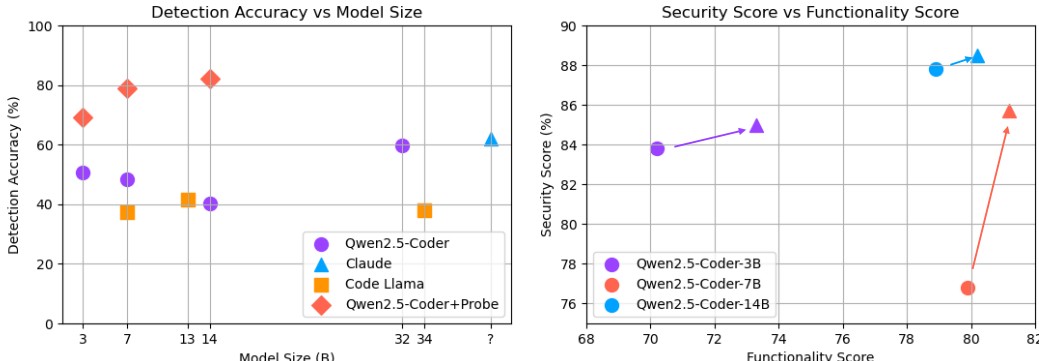

Figure 1: Left: The state-of-the-art code generation models cannot achieve high accuracy by purely prompting, while using probing can improve the accuracy. Right: Security and functional improvements by adding the mixture of corrections.

We integrate these guiding vectors into the model's token generation process, applying conditional corrections with a temporal decay to subtly adjust next-token probabilities based on vulnerability, as assessed from the linear probes. This method enables granular, controlled steering of generation away from vulnerable code while avoiding interference with generation unlikely to yield vulnerable code, and thus without sacrificing functionality. Importantly, we show that it is also possible to apply this process *adversarially* to deliberately increase the likelihood of generating vulnerable code; this may be useful when training future models not to generate vulnerable code.

Our evaluation shows that this conditional steering not only significantly improves the security ratio (8.9% on Qwen2.5-Coder), but also frequently enhances the functional correctness of the resulting code (2.1% on HumanEval) (Figure 1). Moreover, we observe that the guiding vectors sometimes *transfer* across models: vectors derived from one model may improve security in code generated by another model, such as the Qwen-2.5 small-size variants. This transferability yields a computationally efficient way to harden models that are not well-trained specifically on code data.

## 2 RELATED WORK

**LLM-assisted Vulnerability Detection**    Vulnerability detection, a crucial task in computer security, aims to identify potential software security threats, thus reducing the risk of cyber-attacks Lin et al. (2020). LLMs for vulnerability detection use two main paradigms: fine-tuning and prompt engineering Zhou et al. (2025); Shiri Harzevili et al. (2024). Fine-tuning approaches typically introduce a binary classification head on top of the LLM and jointly optimize all model parameters using labeled vulnerable and secure code examples Du et al. (2024). This setup has been applied across various Transformer architectures, including encoder-only Kim et al. (2022); Sun et al. (2023), encoder-decoder Fu et al. (2022), and decoder-only models Zhou et al. (2024b). Some methods Yang et al. (2024a) also use Graph Neural Network backbone to extra features, and concatenate with LLM extracted features. Prompting-based methods Fu et al. (2023) instead query powerful, often proprietary LLMs like GPT-4 using crafted natural language prompts. While these techniques have shown promising results on synthetic datasets Khare et al. (2023), their performance on real-world vulnerability detection tasks is mixed. Recent works have explored structured prompting strategies, such as variations of Chain-of-Thought prompting Ullah et al. (2024), and task-specific prompting frameworks targeting vulnerabilities like Use-Before-Initialization Li et al. (2024) and smart contract bugs Sun et al. (2024). Despite these advances, empirical studies have shown that both fine-tuning and prompting-based methods still struggle with vulnerability detection tasks Ding et al. (2024).

In this work, we propose a new direction: we focus on representation engineering of trained LLMs to improve their internal understanding of secure and vulnerable code; we aim to enhance the latent representations used by the model to reason about code, enabling more robust vulnerability detection without introducing costly retraining or additional inference-time prompt engineering.

**LLM-assisted Secure Code Generation**    The security of the code produced by LLMs remains a pressing concern, prompting a surge of research into security-aware techniques. SafeCoder He et al.

(2024) and ProSec Xu et al. (2024) use security-centric fine-tuning to improve security, utility, and alignment. APILOT Bai et al. (2024) addresses the challenge of outdated or insecure API usage by implementing a mechanism that navigates LLMs to generate secure, version-aware code, thereby reducing potential security threats associated with deprecated APIs. INDICT Le et al. (2024) presents a multi-agent framework that employs internal dialogues between safety-driven and helpfulness-driven critics to iteratively refine code generation, enhancing both the security and functionality of the output. CodeFavor Liu et al. (2024) proposes a code preference model trained on synthetic evolution data to predict whether a code snippet adheres to secure coding practices.

SVEN He & Vechev (2023) is closest to our work; it introduces a method that guides LLMs to generate secure or insecure code by learning a continuous prompt, without modifying the model's weights. This approach allows for controlled code generation based on specified security properties. In contrast to SVEN, we compute a mixture of correction vectors, which are applied conditionally, leading to better control of the code generation. We compare in more detail with SVEN in section 4.

More related works in Steering & Controlling LLM Generation are in subsection A.8.

## 3    A MIXTURE OF LINEAR CORRECTIONS

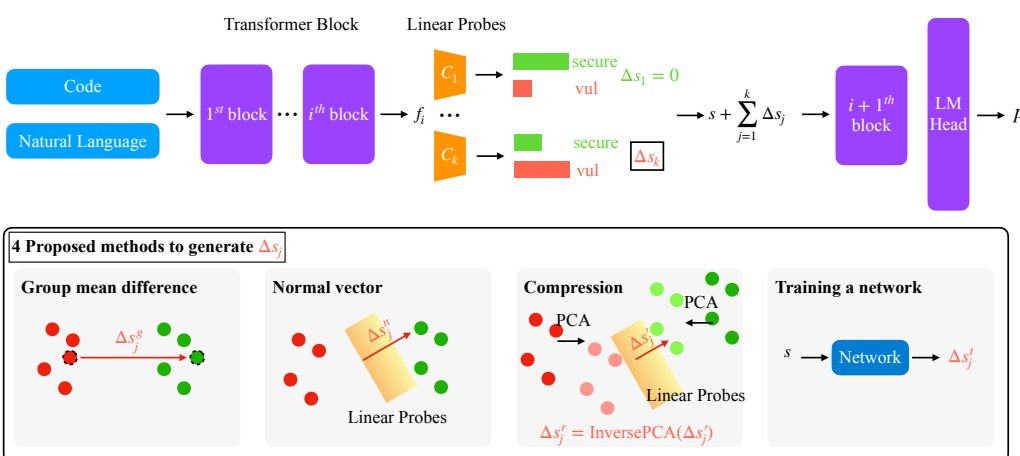

Figure 2: Mixture of corrections (MoC). There are four ways to obtain corrections for each vulnerability $j$. During inference, MoC applies correction $\Delta s_j^{c \in \{g,n,r,t\}}$ if the hidden states are at risk of generating $j^{th}$ type of vulnerable code, and won't make correction if secure.

Figure 2 gives an overview of our approach. We first train a set of linear probes for code vulnerability detection, as described in subsection 3.1, which we then use alongside one of four methods to obtain a set of linear corrections, as described in subsection 3.2. Finally, we use a mixture of the corrections, one for each vulnerability class, to generate secure code, as described in subsection 3.3.

### 3.1    VULNERABLE CODE DETECTION USING LINEAR PROBING

We are given a decoder model used for code generation, denoted $\mathcal{G}$, with transfomer blocks $L_0, \ldots L_n$. We denote the $d$-dimensional activations of the hidden state at the last token position of a block $L_i$ as $s_i$. A *probe* is a diagnostic tool that analyzes the information represented $s_i$, for a particular block $i$. Given a dataset $\mathcal{D}$ of paired (secure, vulnerable) code samples $\{(x^+, x^-)\}$ and a vulnerability type $j$, we will write $\mathcal{D}_j$ to refer to the subset of $\mathcal{D}$ consisting only of vulnerability type $j$.

From the dataset, we use cross entropy loss to train a set of linear probes $c_0(\cdot), \cdots c_k(\cdot)$ against a binary label $v$ which denotes whether the features $s$ were produced by a vulnerable ($x^-$) or secure ($x^+$) sample (Equation 1).

$$\mathcal{L}_c = \text{CE}(c(s), v) = \text{CE}(Ws + b, v) \tag{1}$$

We perform this training on all blocks in the model, and identify the block $L^*$ with the smallest loss to take as the final probe for each vulnerability.

Our empirical investigation reveals that the activations within an LLM exhibit remarkable efficacy for vulnerability detection and the detection accuracy is good compared to previous finetuned or GNN-based detectors. This finding is also a proof that hidden states within LLMs encode richer information than the terminal outputs, as shown in Azaria & Mitchell (2023); Chen et al. (2024), Notably, training these probes only requires minimal data with lightweight parameters.

## 3.2 Controlling Code Generation with a Mixture of Corrections (MoC)

The efficacy of vulnerability detection through representational probing demonstrates that the latent activations within transformer attention mechanisms encode substantive information pertaining to code security vulnerabilities. This observation suggests that these representations can be leveraged to guide code generation—-either to make code more secure, or to deliberately produce vulnerabilities. We propose a framework called *Mixture of Corrections* (MoC) for accomplishing this, which computes a set (mixture) of correction vectors $\Delta s_j$ for each vulnerability class, which are subsequently combined with hidden states when generating code during inference.

We present four methods for computing correction vectors, both static, wherein $\Delta s_j$ is a function only of the vulnerability type $j$, as well as dynamic, where it is also conditioned on the decoder's current hidden states. MoC is illustrated in Figure 2, and detailed in alg. 1.

### 3.2.1 Static Correction Vectors

**Difference of group mean**. The most direct way to measure the correction is by computing the arithmetic differential between the centroid vectors of the respective class data samples as follows:

$$\Delta s_j^g = \frac{1}{|s_j^+|} \sum_{\mathcal{D}_j} s_j^+ - \frac{1}{|s_j^-|} \sum_{\mathcal{D}_j} s_j^- \tag{2}$$

In Equation 2, $|\cdot|$ denotes the cardinality of the set of $j^{th}$ vulnerability.

**Normal vector of the decision boundary**. The linear probe established in the preceding section effectively partitions the feature space into two disjoint subspaces via a linear hyperplane that constitutes the decision boundary. Thus, another way of computing the requisite correction is to traverse orthogonally from the vulnerable class subspace to the non-vulnerable class subspace—specifically, the normal vector to the decision boundary hyperplane. The decision boundary is $(W_{1,:} - W_{0,:})x + b_1 - b_0 = 0$, and the normal vector is characterized in Equation 3.

$$\Delta s_j^n = W_{1,:} - W_{0,:} \tag{3}$$

In Equation 3, $W \in R^{2 \times d}$, and $W_{0,:}$ and $W_{1,:}$ is the first and second row of $W$.

**Reduced normal vector**. Direct utilization of LLM hidden states can exhibit susceptibility to overfitting phenomena and manifest training instability in probe training. Our assumption is that within the high-dimensional feature space, these features encode not only vulnerability-related information but also other types of information that can be considered as noise in code generation contexts. To mitigate these adverse effects, we use dimensionality reduction techniques, specifically principal component analysis (PCA), to derive a more robust correction vector Zou et al. (2023). Let $s_j'$ to denote the compressed version of $s_j$, and we train a linear probe $c_j'(x) = W'x + b'$ on the compressed vectors, then project the correction back to the original high-dimensional space, i.e. $\Delta s_j^r = \text{PCAInverse}(c_j')$, where $W' \in R^{2 \times d'}$, $d'$ is the reduced dimension.

### 3.2.2 Dynamic Correction Vectors

To condition the correction vectors additionally on the dynamic state of the model, for each vulnerability type we train a neural network $N(\cdot)$ that directly predicts the correction vector $\Delta s^t$. To train this network—given that the vulnerability dataset usually contains not only the paired data, but also the detailed line changes of the vulnerable lines of code—we add multiple aspects of supervision. Let $p_j$ denote the $\mathcal{G}$'s output probability corresponding to $s_j$ at the hidden space, and let $y^+$ denote the secure code labels in token space. Note that the $p_j$, $s_j$ and $y^+$ here are per-token supervision, and should be denoted as $p_{j,m}, p_{j,m+1}, \cdots, p_{j,m+n}$, where $m$ is the index for the start of the vulnerable code token, and $m + n$ is the index for the end of the vulnerable code token. We use $p_j$ for simplicity.

The training involves three loss terms, including mean square error, $\mathcal{L}_{mse} = \text{MSE}(s_j^-, s_j^+)$, cross entropy loss $\mathcal{L}_{ce} = \text{CE}(p_j^-, y_j^+)$, and KL-divergence, $\mathcal{L}_{KL} = \text{KL}(p_j^-, p_j^+)$. The final loss form is a combination of these supervisions, $\mathcal{L} = \beta_1 \mathcal{L}_{mse} + \beta_2 \mathcal{L}_{ce} + \beta_3 \mathcal{L}_{KL}$. The network then gives the correction, as $\Delta s_j^t = N(s_j)$.

## 3.3 INFERENCE WITH CORRECTIONS

After obtaining the mixture of corrections $\{\Delta s_j\}$, one for each vulnerability class, we apply the corrections during inference time, by adding the linear combination of corrections to the hidden states when generating every token. However, unlike the previous work Bhattacharjee et al. (2024); Rimsky et al. (2024) which directly adds the vectors, i.e., $f = f + \Delta s$, we find it suboptimal and propose the following improvements.

**Decay**. During inference, as the generation becomes longer, the correction accumulates, which may result in too much correction, and the output generations can be less meaningful. To avoid such a large change in hidden states, we use a decay factor to gradually reduce the impact of the newly added correction during inference.

$$\Delta s := \alpha(t) \cdot \Delta s, \qquad (4)$$

where $t$ is the number of tokens newly generated during inference, e.g., when generating the first token, $t = 0$, and when generating the $k^{th}$ token, $t = k - 1$. $\alpha(\cdot)$ is a negative exponential function.

**Conditional correction**. When generating secure code, there's no need to modify the hidden states and add the correction vectors. The corrections are only applied when the hidden states are at risk of generating vulnerable codes. Thus, we apply a conditional correction, as in the Figure 2, we first use the previously obtained linear probes to detect if the current hidden states are at risk of generating vulnerable codes of vulnerable type $j$, and then apply the corresponding correction $\Delta s_j$ only if the hidden states are vulnerable. If the current hidden states shows multiple different vulnerabilities, then the corrections are added as a linear combination as shown in Equation 5.

$$s+ = \begin{cases} \Delta s_j, & \text{if } \arg\max(c_j(s)) = 0 \\ 0, & \text{otherwise} \end{cases} \qquad (5)$$

We present the overall MoC algorithm in alg. 1. MoC first trains the light-weight linear probes, and then obtains corrections for each type of vulnerability. During inference, MoC applies these corrections if the hidden states are at risk of generating vulnerable code, as measured by the probes.

## 4 EXPERIMENTS

We evaluate the trained probes for vulnerability detection and the mixture of corrections for secure code generation. We also study the effect of MoC on generating functionally correct code and explore the transferability of corrections between models.

### 4.1 VULNERABLE CODE DETECTION

**Dataset**. **Function-level:** Following SVEN He & Vechev (2023), which contributes a high-quality pairwise code dataset for nine different CWEs, we use this dataset as our training and evaluation set. In each vulnerability class, we randomly sample a train set and a subset. Due to the imbalance of the dataset across different types of vulnerabilities, we keep the evaluation set the same size, and the train set might be of different sizes. Notably, the vulnerable and secure data are balanced in our settings. Note that some automatically collected benchmarks Fan et al. (2020); Nikitopoulos et al. (2021) are reported to be inaccurate and less curated Ding et al. (2024). **Repo-level**: We also added experiments on vulnerability detection in more complex and longer contexts, we use PreciseBugs He et al. (2023), which contains much more longer codes (token length up to 30k), with real-world fixes from NVD and OSS-Fuzz provide diverse, authentic bug patterns. Following previous works, we divide the train, validation, and test sets as 80/10/10. And we filter the dataset by filtering out the number of data of a certain CWEs less than 100 in train set. We provide a dataset analysis and examples in the Appendix.

**Evaluation Metric**. Accuracy $\text{Acc}_v$ (%) and $\text{Acc}_s$ (%) is the accuracy of the vulnerable code and secure code, respectively. For training, Acc (%) and F1 (range from 0 to 1) are the accuracy and

---

**Algorithm 1:** Mixture of Corrections

---

**Input:** (1) A code generation LLM $\mathcal{G}$, and its $i^{th}$ transformer blocks $L_i$; (2) A dataset of paired
       vulnerable and secure data $\mathcal{D} = \{\mathcal{D}_j\}$.

**Output:** A secure code generation $x$

   // Training stage

**1 foreach** $j \in \{0, \ldots, k\}$ **do**

**2**      Train the linear probe $c_j$ as in subsection 3.1.

**3**      Obtain correction vector $\Delta s_j^{c \in \{g,n,r,t\}}$ as in subsubsection 3.2.1 & subsubsection 3.2.2.

**4 end**

   // Inference stage

**5 foreach** *token* $x_{t+1}$ **do**

**6**      $s = L^*(x_{1:t})$ ;                           // Get the hidden states

**7**      sum $= 0$ ;

**8**      **foreach** $j \in \{0, \ldots, k\}$ **do**

**9**          **if** $(argmax(c_j(s)) = 0)$ **then**

**10**              sum $=$ sum $+ \alpha(t) \cdot \Delta s_j^m$ ;      // add correction if vulnerable

**11**      **end**

**12**      $s = s +$ sum

**13 end**

**14 return** $x$

---

F1-score on the evaluation set. We also added a manual check for both vulnerability and functionality, as in table 11 and table 12. **Training Details** and **exact hyperparameters** are in the subsection A.7.

**Detection Baselines**: LLMxCPG Lekssays et al. (2025) employs LLMs and code property graphs to first summarize the code and then perform vulnerability detection. VulSim Shimmi et al. (2024) relies on embedding similarity techniques, while VulBERTA Hanif & Maffeis (2022) fine-tunes a RoBERTa model for classification. ReGVD Nguyen et al. (2022) primarily employs graph neural networks to capture structural code relationships for vulnerability identification. LLM-Prompting Steenhoek et al. (2024) uses prompting only, and reports balanced accuracy as the evaluation metric. CodeLlama Roziere et al. (2023) is a foundation model for code generation. LineVul Fu & Tantithamthavorn (2022) analyze each line of code. MSIVD Yang et al. (2024a) merges features from LLM and GNN.

**RQ1: Can LLMs detect vulnerable code by direct prompting?** As evidenced in Table 1, by directly prompting the code generation LLMs, the accuracy is suboptimal. Notably, the prompt includes both few-shot examples sampled from the same dataset (two positive examples and two negative examples), and a description of the specific vulnerability (e.g. CWE-022). We list per-vulnerability experimental results in subsection A.5. We can draw the following conclusions: (1) In general, the current code-related LLMs, including Qwen2.5-Coder series and CodeLlama series, and closed-source model Claude lack the ability to detect code vulnerabilities by prompting. Possible reasons are that the vulnerabilities are less focused on and that these models are not specifically trained on vulnerability code data. (2) There is no clear relation between the model size and their vulnerable detection capacity, though the 32B or 34B models show a small performance improvement compared to smaller models. (3) QC-7B, 14B and 32B, CL-34B models tend to predict the code secure. For the QC-7B, 14B, and CL series, the accuracy is no better than a random guess.

**RQ2: Can hidden states within LLMs help detect vulnerable code?** In Table 3, 'Prompting' means no probe training, and just prompting by few-shot and descriptions as in RQ1. 'Linear Probe W/O Few-shot' refers to, when getting hidden states $f$ from $L_i$ in $\mathcal{G}$, the input only includes the code without few-shot examples. The other probes' input all includes few-shot examples. 'Linear Probe' and 'Linear Probe PCA' contains a linear layer with a weight matrix $W$ and bias $b$, the difference is without PCA $W \in R^{2 \times d}$, where $d = 3168$ in this cases, with PCA, $W' \in R^{2 \times d'}$ and $d'$ is a number between 50 to 100. 'MLP probe' contains 2 or 3 multi-layer perceptron layers, each layer includes a linear layer, a ReLU activation function, a layer norm, and a dropout layer.

From Table 3, Table 2 and Table 5, we can draw the following conclusions. (1) Overall, probing methods can detect vulnerable code, showing that hidden states within LLMs actually contain

Table 1: Accuracy of vulnerable code detection by direct prompting. Invalid means the LLM output doesn't follow the format or includes no answers. QC is short for Qwen2.5-Coder, CL for CodeLlama.

|        | $Acc_v$ | $Acc_s$ | Invalid |
|--------|------|------|---------|
| QC-3B  | 51   | 51   | 0       |
| QC-7B  | 23   | 74   | 3       |
| QC-14B | 25   | 55   | 27      |
| QC-32B | 30   | 81   | 0       |
| CL-7B  | 64   | 10   | 29      |
| CL-13B | 44   | 43   | 14      |
| CL-34B | 21   | 59   | 27      |
| Claude | 63   | 43   | 0       |

Table 2: Comparison of baseline methods and our approach on the SVEN vulnerabilities CWE-125, CWE-190, CWE-416, and CWE-476. Baseline results are taken directly from Lekssays et al. (2025).

|     | VulSim | Vul-BERTA | ReGVD | LLMx CPG |
|-----|--------|-----------|-------|----------|
| Acc | 33     | 50        | 51    | 60       |
| F1  | 31     | 44        | 55    | 70       |

|     | LLM Prompting | Ours QC-14B | Ours QC-14B-PCA |
|-----|---------------|-------------|-----------------|
| Acc | 50.9–54.5     | 70          | 75              |
| F1  | –             | 74          | 77              |

Table 3: Performance on code vulnerability detection; LP denotes Linear Probe.

| Method        | QC-3B | | QC-7B | | QC-14B | |
|---------------|-------|------|-------|------|--------|------|
|               | Acc   | F1   | Acc   | F1   | Acc    | F1   |
| Prompting     | 51    | 0.56 | 49    | 0.53 | 40     | 0.37 |
| LP w/o few-shot | 66  | 0.65 | 68    | 0.66 | 75     | 0.79 |
| LP            | 69    | 0.63 | **79** | **0.76** | **82** | **0.85** |
| LP PCA        | **72** | **0.68** | 76 | 0.74 | 78     | 0.80 |
| MLP Probe     | **72** | 0.66 | 77    | 0.75 | 80     | 0.80 |

Table 4: Latency across different implementations. Upper table uses huggingface; lower table uses vllm.

| Metric   | QC-7B   | MoC          |
|----------|---------|--------------|
| Latency  | 24.56 s | 26.35 s      |
| FLOPS    | 6.9T    | 6.9T + 72M   |
| GPU Mem  | 33 GB   | 33 GB        |
| Latency  | 1.2 s   | 1.8 s        |
| FLOPS    | 6.9T    | 6.9T + 72M   |
| GPU Mem  | 36 GB   | 36 GB        |

vulnerability-related information. From the comparison with the baseline methods, the proposed method shows a clear improvement in vulnerability detection performance. Especially in long repo-level code, the proposed method shows a clear improvement. (2) Using few-shot examples in the text prompt improves the vulnerability detection, showing that the prompting techniques help with the hidden states probing. (3) MLP probes, even with more parameters, don't show a clear improvement compared to linear probes. This may be due to the simplicity of the task: it is a classification task and linear probes are enough to distinguish the secure and vulnerable classes. (4) The performance shows a relation with the LLM scale, as the LLM becomes larger, the performance of the probe gets higher.

Table 5: Detection Results on PreciseBugs. Baseline results are copied from work MSIVD Yang et al. (2024a).

| Technique | F1   | Precision | Recall |
|-----------|------|-----------|--------|
| Random    | 0.29 | 0.20      | 0.50   |
| CodeLlama | 0.22 | 0.16      | 0.35   |
| LineVul   | 0.31 | 0.43      | 0.25   |
| MSIVD     | 0.48 | 0.40      | 0.57   |
| LP        | **0.58** | **0.51** | 0.68 |
| LP PCA    | 0.54 | 0.44      | **0.71** |

## 4.2 SECURE CODE GENERATION

**Benchmark.** Following established practices in prior literature He & Vechev (2023); He et al. (2024), we employ the SVEN test set as our code generation benchmark. While we aimed to incorporate additional repository-level generation benchmarks, we encountered significant limitations in the availability of fully open-source benchmarks specifically designed for repository-level code safety assessment. Although Sec-CodePLT Yang et al. (2024b) represents a potential candidate for such evaluation, we were unable to successfully execute the benchmark due to missing implementation files in the public repository. We will incorporate additional repository-level safe code generation benchmarks as they become available and fully operational in future iterations of this work.

**Evaluation**. We follow the state-of-the-art security evaluation frameworks for LM-based code generators Pearce et al. (2025); He & Vechev (2023); Siddiq & Santos (2022) and evaluate the code

security using GitHub CodeQL CodeQL (2025), an open-source code security analyzer that can detect different vulnerabilities based on the custom queries. We report the security rate SR (%). $\text{SR}_h$ means hardening the security (the higher the better), while $\text{SR}_w$ means weakening the security (the lower the better). The generated code is considered secure only if it doesn't contains any main CWEs based on CodeQL. Note that we test the proposed methods on the SVEN test set, which is different from the evaluation set in subsection 4.1. For code functionality, we test the pass@1 on HumanEval.

**RQ3: Can the mixture of corrections help in secure code generation?** In Table 6, 'Base Model' means applying no corrections. 'SVEN' He & Vechev (2023) is a method that trains prefix soft embeddings and concatenates the embeddings to the LLM during inference. However, on the 3B model, the training loss doesn't decrease, so we choose not to report the results. Then the four correction methods refer to $\Delta s_j^g, \Delta s_j^n, \Delta s_j^r, \Delta s_j^t$ respectively. In the security hardening cases, the conditional generation is utilized, while in the security weakening cases, since the aim is to modify the secure hidden states to insecure ones, and thus it is not conditional, we add the sum of the negative corrections to it, i.e. $\Delta s = \sum_{j=1}^k -\alpha(t)s_j$. In the Table 4, we evaluated on 50 randomly selected test cases from the SVEN dataset using a NVIDIA L40S GPU. The implementation is using huggingface-cli (hf for short) and vLLM. We use `batch_size='auto'` for optimal performance. vLLM achieves significantly faster inference through optimizations like PagedAttention.

As conclusion: (1) Generally, applying MoC can improve not only the security but also the functionality of the code. (2) On Qwen-2.5-Coder-7B, the dynamic NN-based method outperforms others. (3) There are some cases when the weakening cases do not actually bring out more vulnerable codes; one possible reason is that, since we use the sum of all the correction vectors, they may suffer a bit by canceling out on some critical directions. (4) In most cases, the functionality of the LLMs is not affected and even shows some improvements; a possible reason is that the buggy codes have a higher possibility of also being vulnerable Morrison et al. (2015), and there are overlaps between bug-prone code and vulnerabilities Camilo et al. (2015). (5) As for the latency, the lightweight probes have negligible overhead, adding minimal FLOPs, latency, and GPU memory consumption; this makes MoC practical for deployment scenarios where computational efficiency is critical.

Table 6: Performance on code generation. $\text{SR}_h$ (↑)(%) and $\text{SR}_w$ (↓)(%) denote the security ratio when applying hardening and weakening. HE denotes HumanEval pass@1.

| | QC-3B | | | QC-7B | | | QC-14B | | |
|---|---|---|---|---|---|---|---|---|---|
| | $\text{SR}_h$ | $\text{SR}_w$ | HE | $\text{SR}_h$ | $\text{SR}_w$ | HE | $\text{SR}_h$ | $\text{SR}_w$ | HE |
| Base Model | 83.8 | 83.8 | 70.2 | 76.8 | 76.8 | 79.9 | 87.8 | 87.8 | 78.9 |
| SVEN | - | - | - | 65.0 | 54.0 | 75.3 | 69.7 | 65.4 | 75.0 |
| Group Mean Diff | 84.7 | 78.8 | 73.9 | 84.0 | 75.5 | 81.4 | 88.5 | 87.1 | 80.2 |
| Normal Vector | 83.3 | 81.0 | 74.5 | 84.3 | 80.4 | 82.0 | 87.5 | 87.2 | 78.9 |
| Normal Vector PCA | 85.0 | 82.4 | 73.3 | 82.9 | 78.9 | 82.0 | 88.3 | 82.5 | 78.3 |
| Dynamic NN-based | 84.9 | 82.1 | 70.8 | 85.7 | 75.9 | 81.2 | 88.0 | 87.1 | 82.0 |

**RQ4: Which transformer blocks are best for probing?** We train the probes on different transformer blocks, and test their effect on the code generation. As indicated in Figure 3, the last transformer block shows the best performance.

**Ablation Study**. We conduct two versions for PCA corrections. One version is to first obtain both the decision boundary and compute the normal vector to the decision boundary in the compressed space, and then project the normal vector back to the high-dimensional space, as follows:

$$\Delta s_j^r = \text{PCAInverse}(W'_{1,:} - W'_{0,:}) = (W'_{1,:} - W'_{0,:})V + M, \tag{6}$$

where $V$ are the principal components and $M$ are the mean of the vectors. Another is to first project the weighting matrix back and then calculate the normal vector, as follows:

$$\Delta s_j^r = \text{PCAInverse}(W')_{1,:} - \text{PCAInverse}(W')_{0,:} = (W'V)_{1,:} - (W'V)_{0,:} + M_{1,:} - M_{0,:}, \tag{7}$$

note that $W \neq W'V$. We tried both, as in Table 7, the first PCA version fails to generate reasonable outputs, while the second PCA implementation can bring improvements, suggesting that the hidden states space within LLMs is elaborate.

Ablation in Table 8 is conducted on QC-7B model. (1) Adding conditions improves both the secure ratio and the functionality. (2) Though adding decay results in an improvement in the secure ratio, it affects the functionality significantly.

Table 7: Ablations on how to obtain PCA correction.

|  | $SR_h$ | $SR_w$ | HE |
|---|---|---|---|
| Base Model | 76.8 | 76.8 | 79.9 |
| $\Delta s_j^r$ in Equation 6 | 6.3 | 4.6 | 19.8 |
| $\Delta s_j^r$ in Equation 7 | 82.9 | 78.9 | 82.0 |

Table 8: Ablations on conditional correction & decay.

|  | $SR_h$ | HE |
|---|---|---|
| Base Model | 76.8 | 79.9 |
| Normal Vector w/o Condition | 81.7 | 77.0 |
| Normal Vector w/o Decay | 85.8 | 69.6 |
| Normal Vector | 84.3 | 82.0 |

**RQ5: Can the corrections learned for one model transfer to another?** We try to apply the corrections learned from one model and apply them on another model. As in Table 9, the corrections are trained on Qwen2.5-Coder model and implemented on the Qwen2.5-Instruct model, where they share the same hidden dimension and the same model structure. We find that it shows some level of transferability in 3B and 7B models, but not on larger model. However, the functionality of the model based on transferred corrections is harmed on larger models.

Table 9: Transferability across models. We use QI for Qwen2.5-Instruct and QC for Qwen2.5-Coder. HE denotes HumanEval. The corrections are Normal Vector obtained from QC models.

|  | Corrections | $SR_h$ | $SR_w$ | HE |
|---|---|---|---|---|
| QI-7B |  | 76.8 | 76.8 | 69.6 |
| QI-7B | QC-7B | 77.1 | 76.3 | 65.8 |
| QI-3B |  | 75.3 | 75.3 | 54.0 |
| QI-3B | QC-3B | 78.0 | 71.2 | 54.7 |
| QI-14B |  | 63.6 | 63.6 | 74.5 |
| QI-14B | QC-14B | 58.2 | 53.5 | 72.7 |

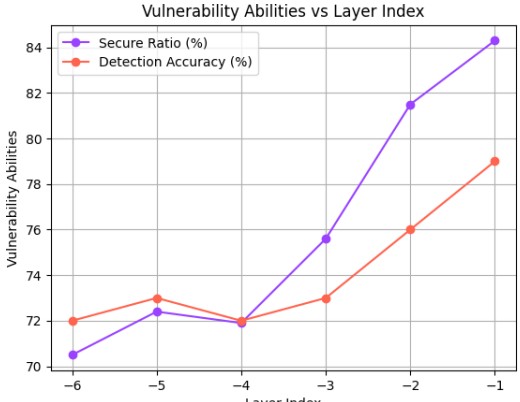

Figure 3: Ablations on $i^{th}$ transformer blocks.

## 5 CONCLUSION

Our investigation reveals that code generation LLMs encode vulnerability-discriminative information in their hidden representations, accessible through lightweight linear probes. We leveraged this insight to develop a Mixture of Linear Corrections (MoC) framework that conditionally applies guiding vectors during inference to enhance code security. Experimental results show our method effectively improves both security ratios and functional correctness across multiple model sizes. This work provides a computationally efficient approach to secure code generation without requiring costly retraining or extensive prompt engineering, opening new avenues for security interventions.

**Limitations.** Currently we use every probe in every token generation, consuming more time than base model; further work can implement acceleration techniques for parallel decoding. While CodeQL serves as our primary evaluation tool, it exhibits inherent limitations in both accuracy and computational efficiency Zhou et al. (2024a). The scarcity of robust automated evaluation methods for code generation tasks presents a significant challenge, with manual human review remaining the most reliable alternative despite its lack of scalability. LLM-as-a-judge is also unsuitable due to its demonstrably poor performance in code vulnerability detection tasks. The development of accurate, scalable evaluation metrics for code generation remains an open research challenge that limits the scope of automated assessment in this domain.

## 6 Ethics Statement

This work adheres to the ICLR Code of Ethics and addresses several ethical considerations relevant to our research. Our study involves automated code generation and vulnerability detection, which raises important concerns about potential misuse. We acknowledge that while our methods are designed to improve software security, the techniques could potentially be adapted for malicious purposes such as generating vulnerable code. To mitigate these risks, we do not release specific vulnerability patterns or exploit generation capabilities. All datasets used in our experiments consist of publicly available code repositories with appropriate licensing, and we have ensured compliance with terms of service for all data sources. We address potential bias in our evaluation by testing across diverse programming languages, with results disaggregated by relevant subgroups reported in the appendix. Privacy concerns are minimal as our work focuses on publicly available code. No human subjects were involved in this research, and no institutional review board approval was required. We declare no conflicts of interest, and all funding sources are acknowledged. Our work aims to contribute positively to software security and developer productivity while being mindful of potential dual-use implications, and we encourage future research to continue developing safeguards against misuse of automated code generation technologies.

## 7 Reproducibility statement

To ensure the reproducibility of our results, we provide comprehensive description of all experimental procedures and resources. Complete source code for our proposed method, including model implementations, training scripts, and evaluation pipelines, is available in the supplementary materials along with detailed installation and usage instructions. All hyperparameters, training configurations, and experimental settings are specified in Appendix A. We report computational requirements, including hardware specifications, training times, and memory usage in Section 4. The supplementary materials also include evaluation scripts and a comprehensive requirements file specifying exact package versions used in our experiments. All datasets used are publicly available with clear citations provided.

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

# A  APPENDIX

## A.1  DATASET DETAILS

We list the full distribution of CWEs included in PreciseBugs as in Tab.10, and this dataset shows some real-world feature: 1) the dataset explicitly reflects the long-tail distribution of security vulnerabilities. For example, CWE-79 contains 7,644 instances, while several other CWEs (e.g., CWE-310, CWE-1333) contain close to 100 instances. This imbalance is included in the dataset. 2), the majority of included CWEs require non-local reasoning over long, multi-file contexts. The average input length is 19,579 tokens (measured using the Qwen-2.5 tokenizer), demonstrating that PreciseBugs captures complex, practical scenarios rather than simplified or toy examples.

**Examples of Safe and Vulnerable Code in PreciseBugs**. Here, we provide two examples and analyze their features, which are (i) no purely local (ii) not trivially caught by static analyzers.

The first example [1] is from PreciseBugs of CWE-269 in Go (actually most of the examples in PreciseBugs contain a full context and represent more than single function) and it's from a github commit, and the second [2]

As shown, they contain multiple files and multiple functions, checking if repos are properly allowed. Actually most cases in PreciseBugs contain more than one function. Also, it's not trivially caught by static analyzers, only if you explicitly encode the business logic into a rule.

Table 10: Distribution of CWEs in PreciseBugs dataset.

| CWE | Count | CWE | Count | CWE | Count |
|-----|-------|-----|-------|-----|-------|
| CWE-79 | 7644 | CWE-918 | 636 | CWE-120 | 296 |
| CWE-89 | 2020 | CWE-863 | 564 | CWE-285 | 146 |
| CWE-352 | 1828 | CWE-190 | 560 | CWE-401 | 148 |
| CWE-125 | 1366 | CWE-400 | 552 | CWE-835 | 144 |
| CWE-200 | 1356 | CWE-269 | 536 | CWE-732 | 132 |
| CWE-20 | 1226 | CWE-74 | 520 | CWE-665 | 132 |
| CWE-787 | 1012 | CWE-94 | 448 | CWE-203 | 128 |
| CWE-119 | 1010 | CWE-601 | 386 | CWE-613 | 116 |
| CWE-476 | 970 | CWE-78 | 380 | CWE-521 | 112 |
| CWE-22 | 944 | CWE-770 | 360 | CWE-330 | 108 |
| CWE-287 | 780 | CWE-434 | 304 | CWE-1333 | 104 |
| CWE-639 | 768 | CWE-416 | 660 | CWE-310 | 102 |

## A.2  PRACTICAL GUIDELINES FOR MOC.

Given that we have explored four different probing and correction method, we summarize, and provide the following takeaways.

1. **Model Scale**: Neural-network–based probes, though with more learnable parameters, tend to perform better at smaller models instead of larger ones, largely because they are less prone to overfitting in this regime.

2. **Computational efficiency**: The group-mean method is the simplest to compute and exhibits strong generalization across settings.

3. **PCA-based methods**: Although PCA-derived vectors—motivated by prior work [Alain Bengio (2017); Zou et al. (2023)]—perform well on the tasks they were originally designed for, they yield comparatively suboptimal performance on our vulnerability-detection task.

---

[1]https://github.com/go-vela/server/commit/05558ee99d70f7d6f83bed7c8f78ac0b35fa26f4.
[2]https://github.com/glpi-project/glpi/commit/aade65b7f67d46f23d276a8acb0df70651c3b1dc

## A.3 MANUALLY CHECK ON VULNERABILITY AND FUNCTIONAL CORRECTNESS

To validate the vulnerability accuracy of CodeQL, we performed a manual analysis, as shown in table 11. For each vulnerability, we select one scenario, then we generate 10 samples and manually assess their vulnerability accuracy with CodeQL results. For example, for instruction in CWE-022, "read a requested file from /safe/", we check whether the function try to access files outside the restricted directory, and compare with CodeQL outputs. As shown in the following table, the CodeQL shows an good accuracy, largely because the test cases in SVEN are well calibrated, hand picked by human and provided with enough contexts.

Table 11: Manually Check on Vulnerability detection for CodeQL.

| CodeQL Acc rate (%) | CodeQL judge as Safe | CodeQL judge as Vulnerable | Language |
|---|---|---|---|
| CWE-022 | 100 | 100 | py |
| CWE-078 | 100 | 100 | py |
| CWE-079 | 100 | 100 | py |
| CWE-089 | 100 | 100 | py |
| CWE-125 | 100 | 100 | c |
| CWE-190 | 100 | 100 | c |
| CWE-416 | 100 | 100 | c |
| CWE-476 | 100 | 100 | c |
| CWE-787 | 100 | 100 | c |

In addition, to validate the generation functionality on SVEN, for each of the nine vulnerabilities in SVEN, we selected one scenario that is straightforward for humans to evaluate (e.g., "read a requested file from /safe/"). For each scenario, we generated 10 samples and manually assessed their functional correctness. We have now added these results as in table 12. As shown in the updated table, the functionality before and after applying MoC remains strong across SVEN, largely because the benchmark focuses on relatively simple functions.

Table 12: Manual Check on SVEN for functionality.

| Acc rate (%) | Before MoC | After MoC | Language |
|---|---|---|---|
| CWE-022 | 100 | 100 | py |
| CWE-078 | 100 | 100 | py |
| CWE-079 | 100 | 100 | py |
| CWE-089 | 100 | 100 | py |
| CWE-125 | 100 | 100 | c |
| CWE-190 | 90 | 90 | c |
| CWE-416 | 100 | 100 | c |
| CWE-476 | 100 | 100 | c |
| CWE-787 | 100 | 100 | c |

## A.4 CROSS-DATASET TRANSFERABILITY

To evaluate MoC's cross-dataset transferability, we trained the probe on SVEN and evaluated it on 100 randomly selected samples from the C and Python subsets of PreciseBugs. The performance is comparable to / a bit better than baseline models, though—as expected—it does not match the results achieved when the model is trained directly on PreciseBugs. It is us some extent applicable to code snippets in the wild, especially at longer code contexts.

## A.5 DETAILED EXPERIMENTAL RESULTS ON DIRECT PROMPTING FOR VULNERABILITY DETECTION.

Here we show the detailed experimental results for each CWEs for each model. The results are from the Table 14 to Table 20, we find that:

| Test on PreciseBugs | F1 | Precision | Recall |
|---|---|---|---|
| SOTA | 0.48 | 0.40 | 0.57 |
| MoC trained on SVEN | 0.49 | 0.42 | 0.59 |
| MoC trained on PreciseBugs | 0.58 | 0.51 | 0.68 |

Table 13: Performance comparison on PreciseBugs dataset

1. Difference CWE types shows every different trends, for example, the CWE-125, CWE-190, CWE-416, CWE-476, CWE-787 contains mostly codes in language c, and Qwen-25-Coder series tend to think they are safe, as in Table 15, Table 16 and Table 17. And the CodeLlama series tend to regard the CWE-416, CWE-476 and CWE-787 as safe, as in Table 19 and Table 20.

2. Overall, QwenCoder series are more recently developed and shows better abilities than CodeLlama series. And overall, the QC series show a better instruction following ability than CL series, as the Invalid rates are lower.

Table 14: Accuracy of vulnerable code detection by direct prompting the LLM. $Invalid_v$ and $Invalid_s$ mean the output of the LLM doesn't follow the format or the output does not include any answers.

| QwenCoder-3B | | | | |
|---|---|---|---|---|
| Vul-type | $Acc_v$ | $Acc_s$ | $Invalid_v$ | $Invalid_s$ |
| 22 | 49 | 49 | 0 | 0 |
| 78 | 51 | 43 | 1 | 0 |
| 79 | 35 | 58 | 0 | 0 |
| 89 | 47 | 61 | 0 | 1 |
| 125 | 52 | 45 | 1 | 0 |
| 190 | 67 | 53 | 0 | 3 |
| 416 | 45 | 62 | 0 | 0 |
| 476 | 59 | 44 | 1 | 1 |
| 787 | 50 | 42 | 0 | 0 |
| Ave | 51 | 51 | 0 | 1 |

Table 15: Accuracy of vulnerable code detection by direct prompting the LLM. $Invalid_v$ and $Invalid_s$ mean the output of the LLM doesn't follow the format or the output does not include any answers.

| QwenCoder-7B | | | | |
|---|---|---|---|---|
| Vul-type | $Acc_v$ | $Acc_s$ | $Invalid_v$ | $Invalid_s$ |
| 22 | 43 | 53 | 0 | 0 |
| 78 | 66 | 72 | 0 | 0 |
| 79 | 44 | 42 | 5 | 5 |
| 89 | 1 | 69 | 0 | 1 |
| 125 | 12 | 77 | 13 | 9 |
| 190 | 8 | 100 | 0 | 0 |
| 416 | 2 | 96 | 0 | 0 |
| 476 | 13 | 84 | 4 | 3 |
| 787 | 15 | 73 | 3 | 2 |
| Ave | 23 | 74 | 3 | 2 |

A.6 DETAILED EXPERIMENTAL RESULTS ON PROBING FOR VULNERABILITY DETECTION.

Here we shows more results about detailed per-CWE results on detection when training a linear probe on the last transformer block. We can draw the conclusion:

Table 16: Accuracy of vulnerable code detection by direct prompting the LLM. Invalid$_v$ and Invalid$_s$ mean the output of the LLM doesn't follow the format or the output does not include any answers.

| | QwenCoder-14B | | | |
|---|---|---|---|---|
| Vul-type | Acc$_v$ | Acc$_s$ | Invalid$_v$ | Invalid$_s$ |
| 22 | 2 | 31 | 69 | 69 |
| 78 | 25 | 62 | 12 | 20 |
| 79 | 28 | 72 | 14 | 14 |
| 89 | 13 | 8 | 78 | 91 |
| 125 | 31 | 64 | 13 | 14 |
| 190 | 36 | 47 | 19 | 11 |
| 416 | 27 | 71 | 16 | 15 |
| 476 | 32 | 74 | 3 | 1 |
| 787 | 35 | 63 | 23 | 6 |
| Ave | 25 | 55 | 27 | 27 |

Table 17: Accuracy of vulnerable code detection by direct prompting the LLM. Invalid$_v$ and Invalid$_s$ mean the output of the LLM doesn't follow the format or the output does not include any answers.

| | QwenCoder-32B | | | |
|---|---|---|---|---|
| Vul-type | Acc$_v$ | Acc$_s$ | Invalid$_v$ | Invalid$_s$ |
| 22 | 43 | 45 | 0 | 0 |
| 78 | 69 | 71 | 0 | 0 |
| 79 | 56 | 44 | 2 | 2 |
| 89 | 99 | 68 | 0 | 1 |
| 125 | 2 | 100 | 0 | 0 |
| 190 | 0 | 100 | 0 | 0 |
| 416 | 0 | 100 | 0 | 0 |
| 476 | 0 | 99 | 0 | 1 |
| 787 | 2 | 100 | 0 | 0 |
| Ave | 30 | 81 | 0 | 0 |

1. Overall, the non-PCA probe in Table 21 shows better results than PCA reduced probes in Table 22. Possible reasons are that the PCA reduced too much information that may be essential for vulnerability detection.

2. Overall, the CWE-022, CWE-078, CWE-079, and CWE-089 are mostly based on python language. And these shows a higher accuracy than other CWEs, especially on QC-14B and 7B models.

3. The CWE-125 and CWE-476 are kind of hard to detect, especially, as the model gets larger, the accuracy on these two CWE-types are not getting higher, which indicates that their vulnerable features are harder to extract.

A.7   MORE DETAILS ABOUT THE IMPLEMENTATION.

**Linear Probe Details.** For each vulnerability, the probe is trained on around 50 to 150 data points due to the imbalance of different types of vulnerabilities. The training epoch is from 50 to 200, with a batch size of 64, learning rate $5e - 4$, SGD optimizer with a momentum and weight decay.

**Training Details.** One of the proposed correction methods requires training of another network, and the network structure is a three-layer multi-layer perception (MLP), with GeLU activation, layer normalization, and dropout layer. The learning rate is $1e - 3$, with an Adam optimizer. To construct the training pair $f_i^+$ and $f_i^-$, we also use the 'line changes' information in each pair of the vulnerable-secure data for detailed supervision. We save the hidden states tensors before training so that training is done on single GPU even for 14B models.

Table 18: Accuracy of vulnerable code detection by direct prompting the LLM. Invalid$_v$ and Invalid$_s$ mean the output of the LLM doesn't follow the format or the output does not include any answers.

| | CodeLlama-7B | | | |
|---|---|---|---|---|
| Vul-type | Acc$_v$ | Acc$_s$ | Invalid$_v$ | Invalid$_s$ |
| 22 | 63 | 0 | 41 | 37 |
| 78 | 49 | 8 | 48 | 48 |
| 79 | 44 | 9 | 49 | 51 |
| 89 | 0 | 0 | 100 | 100 |
| 125 | 89 | 11 | 2 | 2 |
| 190 | 86 | 19 | 3 | 3 |
| 416 | 89 | 9 | 4 | 2 |
| 476 | 59 | 32 | 12 | 12 |
| 787 | 100 | 2 | 0 | 0 |
| Ave | 64 | 10 | 29 | 28 |

Table 19: Accuracy of vulnerable code detection by direct prompting the LLM. Invalid$_v$ and Invalid$_s$ mean the output of the LLM doesn't follow the format or the output does not include any answers.

| | CodeLlama-13B | | | |
|---|---|---|---|---|
| Vul-type | Acc$_v$ | Acc$_s$ | Invalid$_v$ | Invalid$_s$ |
| 22 | 86 | 14 | 4 | 4 |
| 78 | 63 | 29 | 10 | 10 |
| 79 | 47 | 42 | 16 | 21 |
| 89 | 65 | 13 | 26 | 21 |
| 125 | 20 | 18 | 63 | 62 |
| 190 | 83 | 19 | 0 | 0 |
| 416 | 5 | 96 | 2 | 0 |
| 476 | 8 | 72 | 2 | 3 |
| 787 | 20 | 89 | 0 | 1 |
| Ave | 44 | 43 | 14 | 13 |

The exact $\beta_1$, $\beta_2$, $\beta_3$ and $d'$ are 0.2, 1, 0.005, and 64 (for 3B), 80 (7B) and 96 for 14B). For $\beta_3$, it should not be larger than 0.01 and must be used with the gradient norm clipping, otherwise the training will suffer a NaN problem since the KL term can be very large during early stages of training.

Table 20: Accuracy of vulnerable code detection by direct prompting the LLM. Invalid$_v$ and Invalid$_s$ mean the output of the LLM doesn't follow the format or the output does not include any answers.

| | CodeLlama-34B | | | |
|---|---|---|---|---|
| Vul-type | Acc$_v$ | Acc$_s$ | Invalid$_v$ | Invalid$_s$ |
| 22 | 61 | 49 | 8 | 6 |
| 78 | 55 | 37 | 17 | 15 |
| 79 | 7 | 2 | 90 | 90 |
| 89 | 53 | 60 | 2 | 3 |
| 125 | 1 | 54 | 49 | 46 |
| 190 | 0 | 89 | 19 | 17 |
| 416 | 2 | 85 | 15 | 15 |
| 476 | 0 | 81 | 21 | 22 |
| 787 | 10 | 75 | 23 | 19 |
| Ave | 21 | 59 | 27 | 26 |

Table 21: Performance on code vulnerability detection.

| Vul-type | QC-3B | | QC-7B | | QC-14B | |
|---|---|---|---|---|---|---|
| | Acc | F1 | Acc | F1 | Acc | F1 |
| 22 | 0.8 | 0.83 | 0.9 | 0.89 | 0.7 | 0.73 |
| 78 | 0.7 | 0.67 | 0.9 | 0.91 | 0.9 | 0.91 |
| 79 | 0.7 | 0.67 | 0.9 | 0.89 | 0.9 | 0.89 |
| 89 | 0.8 | 0.75 | 0.8 | 0.75 | 1 | 1 |
| 125 | 0.7 | 0.73 | 0.7 | 0.67 | 0.6 | 0.67 |
| 190 | 0.7 | 0.57 | 0.6 | 0.67 | 0.8 | 0.8 |
| 416 | 0.8 | 0.75 | 0.7 | 0.57 | 0.8 | 0.8 |
| 476 | 0.6 | 0.6 | 0.6 | 0.67 | 0.6 | 0.67 |
| 787 | 0.7 | 0.57 | 0.7 | 0.67 | 0.7 | 0.73 |
| Ave | 0.72 | 0.68 | 0.76 | 0.74 | 0.78 | 0.8 |

Table 22: Performance on code vulnerability detection. PCA is applied to the hidden states.

| Vul-type | QC-3B | | QC-7B | | QC-14B | |
|---|---|---|---|---|---|---|
| | Acc | F1 | Acc | F1 | Acc | F1 |
| 22 | 0.6 | 0.33 | 0.8 | 0.75 | 0.8 | 0.8 |
| 78 | 0.7 | 0.57 | 0.9 | 0.89 | 0.9 | 0.91 |
| 79 | 0.6 | 0.5 | 0.9 | 0.91 | 1 | 1 |
| 89 | 0.8 | 0.75 | 0.9 | 0.89 | 1 | 1 |
| 125 | 0.6 | 0.67 | 0.6 | 0.33 | 0.6 | 0.67 |
| 190 | 0.7 | 0.57 | 0.8 | 0.83 | 0.9 | 0.89 |
| 416 | 0.8 | 0.75 | 0.7 | 0.77 | 0.8 | 0.8 |
| 476 | 0.8 | 0.83 | 0.7 | 0.67 | 0.6 | 0.71 |
| 787 | 0.6 | 0.71 | 0.8 | 0.8 | 0.8 | 0.83 |
| Ave | 0.69 | 0.63 | 0.79 | 0.76 | 0.82 | 0.85 |

For $\beta_1$ and $\beta_2$, ranging from 0.1 to 1 doesn't impact final performance, only influences convergence speed.

Table 23: Ablations on concatenation of layers.

| Method | Probe Type | TP/Total | FP/Total | TN/Total | FN/Total |
|---|---|---|---|---|---|
| Activations from last layer | MLP Probe | 0.45 | 0.05 | 0.35 | 0.15 |
| | Linear Probe with PCA | 0.45 | 0.05 | 0.40 | 0.10 |
| Concatenate activations from last two layers | MLP Probe | 0.40 | 0.10 | 0.50 | – |
| | Linear Probe with PCA | 0.45 | 0.05 | 0.40 | 0.10 |
| Concatenate activations from last five layers | MLP Probe | 0.45 | 0.05 | 0.40 | 0.10 |
| | Linear Probe with PCA | 0.45 | 0.05 | 0.45 | 0.05 |

**Prompts**

```
// CWE description
CWE-022, commonly called "Path Traversal," is a vulnerability when an application fails to
appropriately limit . . .
// Few-shot examples
For example, 'code1 . . .' is vulnerable, while 'code2 . . .' is not vulnerable. 'code3 . . .' is
vulnerable, while 'code4 . . .' is not vulnerable.
// Prompt
Is the subsequent code susceptible to the specified vulnerability?
// Test code
code . . .
Answer the question with simply yes or no.
```

### A.8   More Related Works in Steering & Controlling LLM Generation

Controllable generation refers to the ability to steer the outputs of large language models (LLMs) toward desired properties, such as stylistic attributes, factuality, safety, or personalization. A growing body of work has focused on developing techniques for controlling LLMs both at the input and internal representation levels Li et al. (2023); Liang et al. (2024). A prominent strategy for understanding and influencing LLM behavior is probing, which involves training lightweight classifiers on the model's internal activations to extract human-interpretable features Alain & Bengio (2017). Probing has been widely used to reveal latent knowledge in language models and, more recently, to guide and steer generation behavior by identifying representation subspaces associated with specific attributes. Recent advances in representation engineering go beyond passive probing, proposing direct interventions in the model's latent space. These techniques identify semantically meaningful directions in activation space and apply steering vectors to modify model behavior without full retraining Wehner et al. (2025); Zou et al. (2023). Such approaches have been used for tasks like factuality correction, sentiment control, and personalized generation Cao et al. (2024); Wang et al. (2025). Despite this progress, only a few studies He & Vechev (2023); Pimparkhede et al. (2024) have explored the application of controllable generation techniques to code generation, where correctness, determinism, and alignment with developer intent are critical. In this paper, we propose a novel framework that applies probing and representation interventions to code generation models. Our method performs conditional interventions in the activation space, guided by the outputs of probes trained to detect semantic properties or vulnerabilities in the code.

### A.9   Boarder Impact

Our work on code vulnerability detection and correction has significant societal implications. **Positively**, it enhances code security, potentially reducing data breaches and cyberattacks that impact millions of users annually. Secure code generation tools democratize cybersecurity expertise, benefiting resource-constrained organizations and critical infrastructure. **Negatively**, adversarial applications of our techniques could be used to deliberately introduce subtle vulnerabilities or automate exploitation of existing weaknesses. Additionally, over-reliance on automated security tools may create false confidence and reduce human oversight. We encourage responsible deployment with human-in-the-loop verification and recommend against using these methods in high-risk applications without thorough testing.

## B   Use of Large Language Models

Large Language Models were used as an assistive tool during the preparation of this manuscript. Specifically, we employed LLMs (including but not limited to GPT-based models and Claude) for rephrasing and improving the clarity of certain sections of the paper, particularly in refining sentence structure, enhancing readability, and ensuring consistent academic writing style throughout the manuscript. The LLMs were used to rephrase existing content that was originally authored by the research team, rather than for generating novel research ideas, methodological approaches, or substantive technical content. All core research contributions, experimental design, analysis, and

conclusions are the original work of the authors. The LLMs did not participate in research ideation, hypothesis formation, or the interpretation of results. The use of LLMs was limited to language assistance and did not influence the scientific validity or originality of our work.

