# OpenReview forum: "A Mixture of Linear Corrections Generates Secure Code"
_ICLR.cc/2026/Conference — Submitted to ICLR 2026_

### Official Review · Reviewer_bWZM · 2025-10-26

**Soundness:** 3
**Presentation:** 3
**Contribution:** 2
**Rating:** 4
**Confidence:** 5

**Summary:**

The paper empirically shows that, from some common CWEs, large code models possess latent representations that distinguishes vulnerable code from secure code, even if they can’t reliably explain it when prompted: simple linear probes trained on hidden activations can classify vulnerable vs fixed code across specific CWEs with much higher accuracy than direct prompting and even beat prior automated vulnerability detectors on both function snippets and full real-world repos, meaning vulnerability information is already linearly separable in the model’s internal state. Building on that, the authors introduce Mixture of Corrections (MoC), an inference-time activation-steering method: for each CWE, they learn a correction direction that shifts the model’s hidden state from “vulnerable-like” toward “secure-like,” then at generation time they (1) detect when the current hidden state looks risky using the probe and (2) add the relevant correction vectors (with decay over time), leaving the model alone when it’s already on a safe path. This conditional steering improves both security and functionality: for example, on Qwen2.5-Coder-7B, the share of generations with no CodeQL-detected vulnerabilities increases by nearly 9 percentage points and HumanEval pass@1 also goes up by about 2 points, which is notable because most “secure coding” interventions hurt correctness. The approach is lightweight (a few extra vector ops per token, no full fine-tune), sometimes even transfers across related model variants, and can also be inverted to intentionally induce vulnerabilities (useful for red-teaming), though the authors acknowledge dual-use risk and limits of static analyzers like CodeQL for ground truth.

**Strengths:**

- The paper convincingly shows that vulnerability information is already encoded in the internal activations of code LLMs (**for certain CWE types covered in the experiments**), because simple linear probes can reliably distinguish vulnerable vs fixed code and even outperform prior static/dynamic vulnerability detection systems on both function-level and repo-level datasets.
- The paper proposed a practical steering approach MoC, leveraging the observations, to intervene the generation process and make code LMs generate safer code, which is light-weights and has certain interpretability compared to existing solutions.
- The proposed approach improve the security of generated code without harming the utility of code LMs.
- The paper shows that in some cases steering vectors trained on one model can be applied to a related model and still improve security, which hints at reuse across a model family and lowers the cost of adoption.

**Weaknesses:**

- The method works per CWE type, which is a strength in targeting, but also a scaling bottleneck. Each new vulnerability class needs labeled vulnerable/fixed pairs, a trained probe, and a steering vector.
- Real-world code has long-tail, app-specific logic flaws (auth bypass, race conditions, crypto misuse) that may not map cleanly to the benchmark CWEs the paper evaluates. The paper doesn’t fully address how this would generalize beyond the covered CWEs.
- The experiment setting seems to have removed CWE categories with training samples less than 100 (line 266-267). This will remove most of the hard CWEs that are actually the critical part of research to push the boundary of secure code generation, compared to the relatively simple ones that can be fixed by adding/modifying a line, e.g., substituting an unsafe serialization API with a safe one. Note that this is orthogonal to length of the code. Missing complex types of CWE will dramatically influence the overall experiment results and cause bias (**potential overclaiming**) in the conclusion that LLMs possess knowledge in differentiating vulnerable code and safe code.
- The work argues MoC is practical for “real devs,” but we don’t see a user study measuring whether the outputs are actually easier to review, have fewer subtle logic bugs, or reduce time-to-fix.
- The MoC approach is intrinsically similar to some MoE-style adapters. Maybe some kind of baselines need to be added here.

**Questions:**

1. Can you provide a deeper analysis on the vulnerability detection and steering ability of MoC on different vulnerability types, especially those fundamental different from the frequent ones, e.g. (i) not purely local and (ii) not trivially caught by static analyzers? In particular, can you include a CWE that depends on multi-function, rather than a single usage site?
2. Since the steering approach proposed in this work falls into a broader domain of lightweight adaptation of LMs, can you add some comparison with existing parameter efficient finetuning approaches (especially MoE-style) under similar settings and discuss their relationship, e.g. [1,2].

> [1] Wang, Renzhi, and Piji Li. "Lemoe: Advanced mixture of experts adaptor for lifelong model editing of large language models." arXiv preprint arXiv:2406.20030 (2024).\
> [2] Wu, Xun, Shaohan Huang, and Furu Wei. "Mixture of LoRA Experts." The Twelfth International Conference on Learning Representations.

---

> ### Author Response · Authors · 2025-11-18
> **Response to Reviewer bWZM 1/2**
>
> Thank you for your time and your valuable review.
>
> **Weakness 1**: *The method works per CWE type, which is a strength in targeting, but also a scaling bottleneck. Each new vulnerability class needs labeled vulnerable/fixed pairs, a trained probe, and a steering vector.*
>
> **Response**: Yes, each probe and correction module in our method is trained specifically for a single CWE. We adopt this design for two main reasons:
>
> 1. **General steering vectors are ineffective for CWE control.** Prior work on steering vectors (e.g., sentiment  style transfer or general safety steering) has shown limited effectiveness, largely because such vectors are overly general and the underlying latent space is highly entangled. Actually, we attempted to learn a single shared vector for nine CWEs, but its performance was significantly worse than using CWE-specific vectors. This empirical result supports our choice to train dedicated probes and corrections for each CWE.
>
> 2. **CWE definitions are dynamic and vary in practical importance.** New CWEs continue to be introduced as software engineers identify additional vulnerability patterns, and certain CWEs pose substantially greater risk than others. Our approach allows practitioners to prioritize the CWEs most relevant to their applications and to easily add new or high-impact CWEs when needed. In this sense, MoC provides practical flexibility rather than imposing a fixed or static set of controls.
>
> We hope this clarifies why CWE-specific training is both empirically justified and practically advantageous.
>
> **Weakness 2**: *Real-world code has long-tail, app-specific logic flaws (auth bypass, race conditions, crypto misuse) that may not map cleanly to the benchmark CWEs the paper evaluates. The paper doesn’t fully address how this would generalize beyond the covered CWEs.*
>
> **Response**: The PreciseBugs benchmark already incorporates substantial real-world diversity and practical challenges.
>
> 1. **The dataset explicitly reflects the long-tail distribution of security vulnerabilities.** For example, CWE-79 contains 7,644 instances, while several other CWEs (e.g., CWE-310, CWE-1333) contain close to 100 instances. This imbalance is included in the dataset.
>
> 2. **The majority of included CWEs require non-local reasoning over long, multi-file contexts**. The average input length is 19,579 tokens (measured using the Qwen-2.5 tokenizer), demonstrating that PreciseBugs captures complex, practical scenarios rather than simplified or toy examples.
>
> For completeness, we list the full distribution of CWEs included in PreciseBugs:
>
> {
>  ‘CWE-79’: 7644, ‘CWE-89’: 2020, ‘CWE-352’: 1828, ‘CWE-125’: 1366, ‘CWE-200’: 1356, ‘CWE-20’: 1226,
>  ‘CWE-787’: 1012, ‘CWE-119’: 1010, ‘CWE-476’: 970, ‘CWE-22’: 944, ‘CWE-918’: 636, ‘CWE-863’: 564,
>  ‘CWE-190’: 560, ‘CWE-400’: 552, ‘CWE-74’: 520, ‘CWE-94’: 448, ‘CWE-601’: 386, ‘CWE-78’: 380,
>  ‘CWE-770’: 360, ‘CWE-434’: 304, ‘CWE-120’: 296, ‘CWE-269’: 536, ‘CWE-416’: 660, ‘CWE-287’: 780,
>  ‘CWE-639’: 768, ‘CWE-521’: 112, ‘CWE-732’: 132, ‘CWE-203’: 128, ‘CWE-613’: 116, ‘CWE-285’: 146,
>  ‘CWE-835’: 144, ‘CWE-665’: 132, ‘CWE-330’: 108, ‘CWE-1333’: 104, ‘CWE-310’: 102, ‘CWE-401’: 148
>  }
>
> We hope this clarifies that PreciseBugs is designed to faithfully represent practical, real-world bug detection challenges.
>
> **Weakness 3**: *The experiment setting seems to have removed CWE categories with training samples less than 100 (line 266-267). This will remove most of the hard CWEs that are actually the critical part of research to push the boundary of secure code generation, compared to the relatively simple ones that can be fixed by adding/modifying a line, e.g., substituting an unsafe serialization API with a safe one. Note that this is orthogonal to length of the code. Missing complex types of CWE will dramatically influence the overall experiment results and cause bias (potential overclaiming) in the conclusion that LLMs possess knowledge in differentiating vulnerable code and safe code.*
>
> **Response**: Our intention was not to exclude any challenging CWEs; the selection is solely constrained by what is available in the public datasets we use. Compared with existing datasets, ours already incorporates a broad range of realistic and complex vulnerability types, and we view this as a meaningful step toward comprehensive coverage.
>
> For CWEs that are particularly important but underrepresented, we believe a practical remedy is to synthesize additional samples for the specific CWE and train a corresponding probe using MoC. We have also tried to train on 50 samples and test on 20 samples, with results reported below. Note that it’s comparable to the results of LLMxCPG, which tunes and uses two 32B models. The results show that with less examples on SVEN using QwenCoder7B MoC still performs reasonable.
>
> | On SVEN with QC-7B | F1 | Acc |
> |--------------------|----|----|
> | CWE-125            | 70 | 60  |
> | CWE-190            | 70 | 65  |

---

> ### Author Response · Authors · 2025-11-18
> **Response to Reviewer bWZM 2/2**
>
> -----continue------
>
> **Weakness 4**: *The work argues MoC is practical for “real devs,” but we don’t see a user study measuring whether the outputs are actually easier to review, have fewer subtle logic bugs, or reduce time-to-fix.*
>
> **Response**: We claim MoC is practical because it is lightweight,  trains efficiently, and supports flexible integration of new vulnerabilities.
>
> Regarding concerns about logical or functional correctness, we note that we have already reported results on HumanEval, a standard benchmark for assessing code functionality. MoC demonstrates improvements in most cases in this benchmark.
>
> We do not claim that MoC makes code easier to review or reduces the time required to fix issues; therefore, we believe empirical studies targeting those dimensions are outside the scope of this work.
>
> **Weakness 5 and Question 2**: *The MoC approach is intrinsically similar to some MoE-style adapters. Maybe some kind of baselines need to be added here. Since the steering approach proposed in this work falls into a broader domain of lightweight adaptation of LMs, can you add some comparison with existing parameter efficient finetuning approaches (especially MoE-style) under similar settings and discuss their relationship, e.g. [1,2].*
>
> **Response**: Sure, we would love to discuss the differences in related works.
>
> Methods such as [1,2] offer efficient and widely used mechanisms for incorporating knowledge or improving LLM performance on specific tasks. These approaches also hold promise for vulnerability detection and safe code generation. However, they typically require additional data when training each expert module, which has the potential to perform better in data-sufficient settings.
>
> **Question 1**: *Can you provide a deeper analysis on the vulnerability detection and steering ability of MoC on different vulnerability types, especially those fundamental different from the frequent ones, e.g. (i) not purely local and (ii) not trivially caught by static analyzers? In particular, can you include a CWE that depends on multi-function, rather than a single usage site?*
>
> **Response**: An example (i) no purely local (ii) not trivially caught by static analyzers:
>
> This is from PreciseBugs of CWE-269 in Go (actually most of the examples in PreciseBugs contain a full context and represent more than single function) and it’s from a github commit here:
> https://github.com/go-vela/server/commit/05558ee99d70f7d6f83bed7c8f78ac0b35fa26f4.
>
> 1. Not local: as shown, it contains two functions, checking if repos are properly allowed. Actually most cases in PreciseBugs contain more than one functions.
> 2. Also, it’s not trivially caught by static analyzers, only if you explicitly encode the business logic into a rule.
>
> Here is another example This is from PreciseBugs of CWE-639 in php, and it’s from a github commit here: https://github.com/glpi-project/glpi/commit/aade65b7f67d46f23d276a8acb0df70651c3b1dc. We can see that it contains multiple files, and CWE-639 is the type that’s very unlikely detected by static analyzer.
>
> MoC can successfully detect this vulnerability, while other methods, e.g. sota method, which trains a GNN and an LLM and concatenate the features, can not. Note that the sota method doesn’t open source their code, and do not have separate results on each CWE, so we report their average F1. The table below shows that MoC performs better than baselines on these complex CWEs.
>
> |    | Sota (on average) | MoC CWE-269 | MoC CWE-639 |
> |----|-------------------|-------------|-------------|
> | F1 | 0.48              | 0.56        | 0.58        |
>
>
> Thank you again for your time and for the valuable feedback. Please let us know if any points require further clarification—we would be glad to provide additional details.

---

> > ### Comment · Reviewer_bWZM · 2025-11-26
> >
> > My concerns are addressed. I'll raise my score.

---

> > > ### Author Response · Authors · 2025-11-26
> > >
> > > Thanks for the constructive feedback throughout the review process. We appreciate your time and thoughtful
> > > engagement with our work. We will incorporate all the discussed improvements in the updated version as promised.

---

### Official Review · Reviewer_6d6p · 2025-10-28

**Soundness:** 3
**Presentation:** 3
**Contribution:** 3
**Rating:** 4
**Confidence:** 3

**Summary:**

This paper leverages representation techniques and inference-time steering methods to enhance large language models’ capabilities in bug detection and secure code generation. It employs linear probing on hidden states to detect vulnerabilities and propose four types of correction vectors: the Difference of Group Means, the Normal Vector, the PCA-Reduced Normal Vector, and NN-based Dynamic Corrections.

The evaluation shows that compared with prompt-based methods, linear probing improves detection accuracy. In secure code generation, the proposed Mixture of Corrections (MoC) increases the security rate without compromising functionality.

**Strengths:**

- Employs linear probing on hidden representations, achieving better bug detection accuracy than prompt-based baselines.
- Introduces a Mixture of Corrections that improve code security while maintaining functionality.
- Demonstrates that the learned correction vectors exhibit a certain degree of cross-model transferability.

**Weaknesses:**

- Although four types of corrections are proposed, the paper does not clearly describe how they are combined for a given bug type. Are all four used simultaneously, or is only one applied each time?
- Each correction is trained specifically for one bug type (CWE). This means for multiple bug types, separate probes and corrections must be trained, potentially increasing computational overhead and raising questions about interaction or interference between corrections when multiple vulnerabilities coexist.

**Questions:**

* During inference, how are the different corrections mixed? Is only one type applied at a time, or are they linearly combined?
* In Table 6, the performance gap between different correction methods is relatively small. What are the practical advantages of introducing PCA-based or NN-based dynamic corrections? Could the authors provide guidelines on when to prefer each type?
* Besides vulnerability type, are the trained corrections also language-dependent? If so, would this require retraining separate probes and corrections for each (CWE type, programming language) pair?

---

> ### Author Response · Authors · 2025-11-18
> **Response to Reviewer 6d6p**
>
> Thank you for your time and your valuable review.
>
> **Weakness 1 and question 1**: *Although four types of corrections are proposed, the paper does not clearly describe how they are combined for a given bug type. Are all four used simultaneously, or is only one applied each time? During inference, how are the different corrections mixed? Is only one type applied at a time, or are they linearly combined?*
>
> **Response**: Yes, only a single correction method is applied in each experiment. The four correction strategies are independent of one another, and we evaluate them individually rather than in combination.
>
> **Weakness 2**: *Each correction is trained specifically for one bug type (CWE). This means for multiple bug types, separate probes and corrections must be trained, potentially increasing computational overhead and raising questions about interaction or interference between corrections when multiple vulnerabilities coexist.*
>
> **Response**: Yes, each probe and correction module in our method is trained specifically for a single CWE. We adopt this design for two main reasons:
>
> 1. **General steering vectors are ineffective for CWE control.**  Prior work on steering vectors (e.g., sentiment style transfer or general safety steering) has shown limited effectiveness, largely because such vectors are overly general and the underlying latent space is highly entangled. Actually, we attempted to learn a single shared vector for nine CWEs, but its performance was significantly worse than using CWE-specific vectors. This empirical result supports our choice to train dedicated probes and corrections for each CWE.
>
>
> 2. **CWE definitions are dynamic and vary in practical importance.** New CWEs continue to be introduced as software engineers identify additional vulnerability patterns, and certain CWEs pose substantially greater risk than others. Our approach allows practitioners to prioritize the CWEs most relevant to their applications and to easily add new or high-impact CWEs when needed. In this sense, MoC provides practical flexibility rather than imposing a fixed or static set of controls.
>
> We hope this clarifies why CWE-specific training is both empirically justified and practically advantageous.
>
> **Question 2**: *In Table 6, the performance gap between different correction methods is relatively small. What are the practical advantages of introducing PCA-based or NN-based dynamic corrections? Could the authors provide guidelines on when to prefer each type?*
>
> **Response**: Yes, different correction methods yield not-so-different performance. Because our work focuses on analyzing and applying correction vectors, we experimented with several options to ensure robustness. Based on our observations, we offer the following clarifications and practical guidelines:
>
> 1. Model Scale: Neural-network–based probes, though with more learnable parameters, tend to perform better at smaller models instead of larger ones, largely because they are less prone to overfitting in this regime.
>
> 2. Computational efficiency: The group-mean method is the simplest to compute and exhibits strong generalization across settings.
>
> 3. PCA-based methods: Although PCA-derived vectors—motivated by prior work [Alain & Bengio (2017); Zou et al. (2023)]—perform well on the tasks they were originally designed for, they yield comparatively suboptimal performance on our vulnerability-detection task.
>
> We will add these empirical guidelines to the revised version of the paper.
>
> **Question 3**: *Besides vulnerability type, are the trained corrections also language-dependent? If so, would this require retraining separate probes and corrections for each (CWE type, programming language) pair?*
>
> **Response**: We appreciate the reviewer’s comment. In SVEN, the languages are C and Python, while PreciseBugs covers C, C++, Rust, Go, and JVM languages. For these datasets, the learned corrections are not inherently language-dependent. However, to the best of our knowledge, vulnerability patterns can be language-dependent or language-influenced. Certain CWEs occur far more frequently in one language than another (e.g., in SVEN, some CWEs appear exclusively in a single language).
>
> Thus, for the purposes of this paper, we state that the corrections we study are not strongly dependent on the programming language and often follow similar remediation patterns (e.g., cwe-022: properly restricting or sanitizing user-supplied pathnames). Nevertheless, we agree that, in general, both CWE distributions and correction strategies are likely to benefit from language-specific training.
>
>
>
> Thank you again for your time and for the valuable feedback. Please let us know if any points require further clarification—we would be glad to provide additional details.

---

### Official Review · Reviewer_dtL1 · 2025-10-30

**Soundness:** 3
**Presentation:** 3
**Contribution:** 2
**Rating:** 4
**Confidence:** 4

**Summary:**

The paper studies the problem of vulnerability detection and secure code generation using LLMs. The first key claim is that linear probes over internal features of LLMs can be used to detect vulnerable code. Moreover, these features can be steered linearly to generate secure code. A Mixture of Corrections (MoC) method is proposed to steer a feature in the direction of a collection of "correction vectors" (one for each vulnerability type) -- the correction vector for a vulnerability type is included if the associated linear probe detects the feature as vulnerable. Experiments show that linear probing outperforms neural detection baselines and standard prompting with LLMs on vulnerability detection. Further, MoC improves the security ratio for three Qwen-x models (and reduces it in the security weakening setup).

**Strengths:**

- The paper studies the important problems of vulnerability detection and secure code generation with LLMs.
- The paper is well-written and the key ideas are easy to understand.
- The use of linear probing to detect vulnerabilities is novel.

**Weaknesses:**

- Some of the steering methods considered in Section 3.2.1 have been proposed for other natural language tasks (difference of group mean [1], normal vector of the decision boundary [2]).
- The secure code generation task reports the security ratio metric but does not report the correctness of the outputs after steering with MoC on the SVEN Test Set (Table 6). There could potentially be a trade-off between the security ratio and the correctness of generation after steering (similar to the accuracy fluency tradeoff observed by other steering works [3]). At the least, I would expect a short manual analysis of the generated outputs with a discussion on the quality / correctness of the generated secure code.
- The expeirments do not include a "control" benchmark from an unrelated task, i.e., a benchmark with no vulnerabilities. Ideally, this should be used to demonstrate that MoC has no adverse affect on prompts / examples which do not include any vulnerabilities (and were not included in training the linear probe).

Overall, I think that the novelty in terms of techniques used is limited (albeit novel in the context of vulnerability detection and secure code generation). The steering experiments would benefit from some qualitative examples of the generated code and reporting the accuracy of the correctness of the generated secure code. The detection experiments would benefit from the inclusion of a control benchmark.

[1] Ole Jorgensen, Dylan Cope, Nandi Schoots, and Murray Shanahan. Improving activation steering in language models with mean-centring. arXiv preprint arXiv:2312.03813, 2023.

[2] Kenneth Li, Oam Patel, Fernanda Viégas, Hanspeter Pfister, and Martin Wattenberg. Inference-time intervention: Eliciting truthful answers from a language model. Advances in Neural Information Processing Systems, 36: 41451–41530, 2023.

[3] Guardieiro, V., Stein, A., Khare, A., & Wong, E. (2025). Instruction Following by Boosting Attention of Large Language Models. Mechanistic Interpretability Workshop at NeurIPS 2025. Retrieved from https://openreview.net/forum?id=xDyJVMnab8.

**Questions:**

I will summarize my questions from the weaknesses section above (please refer to that section for more details):
1. How are the proposed steering methods (the group mean and the normal vector) different from those proposed in [1] and [2]?
2. Could you share some qualitative examples of the outputs after steering?
3. Are the probes trained and tested on splits from the same dataset? If yes, how would the results change if the probe is trained on SVEN and tested on PreciseBugs? This is important to know whether this can be applicable to code snippets in the wild (i.e., not from a specific dataset).
4. Are the probes and steering vectors trained on hidden states from the residual layer (after a transformer block) or attention blocks? RQ4 mentions attention blocks while Section 3.1 mentions transformer blocks.

---

> ### Author Response · Authors · 2025-11-18
> **Response to Reviewer dtL1 1/2**
>
> Thank you for your time and your valuable review.
>
> **Weakness 1 and question 1**: *Some of the steering methods considered in Section 3.2.1 have been proposed for other natural language tasks (difference of group mean [1], normal vector of the decision boundary [2]).How are the proposed steering methods (the group mean and the normal vector) different from those proposed in [1] and [2]?*
>
> **Response**: While steering has indeed been explored in much of the literature, it has not been widely adopted in industry and is often regarded as less effective across many tasks, such as sentiment style transfer, general safety rewriting, etc.
>
> Interestingly, in our setting, it is consistently beneficial for code generation. We believe the key difference lies in MoC’s usage: rather than training a generic probe and using it regardless, we construct a dedicated steering vector for each specific vulnerability and apply it in a targeted manner. Our contribution is therefore not the introduction of a new steering technique, but a demonstration of how and when steering can be used effectively in this domain.
>
> **Weakness 2 and question 2**: *The secure code generation task reports the security ratio metric but does not report the correctness of the outputs after steering with MoC on the SVEN Test Set (Table 6). There could potentially be a trade-off between the security ratio and the correctness of generation after steering (similar to the accuracy fluency tradeoff observed by other steering works [3]). At the least, I would expect a short manual analysis of the generated outputs with a discussion on the quality / correctness of the generated secure code. Could you share some qualitative examples of the outputs after steering?*
>
> **Response**: Yes, the original SVEN benchmark does not include an explicit metric for output correctness. Thus, we report HumanEval functionality results in the paper—which indicate that MoC offers modest improvements in most cases.
>
> We appreciate your suggestion and thus have conducted an additional analysis to more directly address this point. Specifically, for each of the nine vulnerabilities in SVEN, we selected one scenario that is straightforward for humans to evaluate (e.g., “read a requested file from /safe/”). For each scenario, we generated 10 samples and manually assessed their functional correctness. We have now added these results to the paper. As shown in the updated table, the functionality before and after applying MoC remains strong across SVEN, largely because the benchmark focuses on relatively simple functions.
>
> | Acc rate (%) | Before MoC | After MoC | Language |
> |--------------|------------|-----------|----------|
> | CWE-022      | 100        | 100       | py       |
> | CWE-078      | 100        | 100       | py       |
> | CWE-079      | 100        | 100       | py       |
> | CWE-089      | 100        | 100       | py       |
> | CWE-125      | 100        | 100       | c        |
> | CWE-190      | 90         | 90        | c        |
> | CWE-416      | 100        | 100       | c        |
> | CWE-476      | 100        | 100       | c        |
> | CWE-787      | 100        | 100       | c        |
>
> **Weakness 3**: *The experiments do not include a "control" benchmark from an unrelated task, i.e., a benchmark with no vulnerabilities. Ideally, this should be used to demonstrate that MoC has no adverse effect on prompts / examples which do not include any vulnerabilities (and were not included in training the linear probe).*
>
> **Response**: We would like to clarify that we do report results on benchmark with no vulnerabilities. We evaluate HumanEval, a benchmark that evaluates general code functionality. These results demonstrate that, in most cases, our proposed method improves the model’s overall functional performance. We believe this occurs because a steering vector trained on safe code not only reduces insecure patterns but also captures characteristics of high-quality code, thereby providing a broader benefit to general coding ability.

---

> ### Author Response · Authors · 2025-11-18
> **Response to Reviewer dtL1 2/2**
>
> ------continue------
>
> **Question 3**: *Are the probes trained and tested on splits from the same dataset? If yes, how would the results change if the probe is trained on SVEN and tested on PreciseBugs? This is important to know whether this can be applicable to code snippets in the wild (i.e., not from a specific dataset).*
>
> **Response**: Yes, the probe is trained and evaluated on splits derived from the same dataset. We would also like to clarify that SVEN and PreciseBugs are fundamentally different benchmarks: SVEN is function-level (average length ≈ 800 tokens per training sample, languages: C and Python), whereas PreciseBugs is repository-level (average length ≈ 20,000 tokens per training sample, languages: C, Python, Go, JVM, Rust).
>
> To answer your question, we trained the probe on SVEN and evaluated it on 100 randomly selected samples from the C and Python subsets of PreciseBugs. The performance is comparable to / a bit better than baseline models, though—as expected—it does not match the results achieved when the model is trained directly on PreciseBugs. It shows that MoC is to some extent applicable to code snippets in the wild, especially at longer code completions shows better results than other baselines.
>
> | Test on PreciseBugs | F1 | Precision | Recall |
> |---------------------|-----|-----------|--------|
> | SOTA | 0.48 | 0.40 | 0.57 |
> | MoC trained on SVEN | 0.49 | 0.42 | 0.59 |
> | MoC trained on PreciseBugs | 0.58 | 0.51 | 0.68 |
>
> **Question 4**: *Are the probes and steering vectors trained on hidden states from the residual layer (after a transformer block) or attention blocks? RQ4 mentions attention blocks while Section 3.1 mentions transformer blocks.*
>
> **Response**: The probes are trained after a transformer block; we will update the wording correspondingly.
>
> Thank you again for your time and for the valuable feedback. Please let us know if any points require further clarification—we would be glad to provide additional details.

---

> ### Comment · Reviewer_dtL1 · 2025-11-23
>
> Thank you for the response. I appreciate the time and effort that the authors put into answering the questions and reporting results from the additional analyses. I would appreciate some comments on some follow-up questions:
>
> **Question 1:** Since the steering techniques have been proposed in prior works, I would encourage the authors to clearly mention this (and cite the corresponding works) when discussing the techniques in Section 3.2.1.
>
> **Question 2:** Thank you for the additional analysis! Could you please point me to the section in the paper with these new results? I would like to see some qualitative examples of the steered generations.
>
> Also, since you mention that the functional correctness is expected because of the benchmark being simple, could you also share some examples from PreciseBugs? I understand that evaluation of repo-level samples is difficult but a few qualitative examples of the changes made to the code by MoC would suffice.
>
> **Question 3:** Thank you for the ablation. My interpretation of these results is that MoC with probes trained on SVEN are only marginally better than the baseline models on PreciseBugs. Hence, I don't agree with the statement that this shows that MoC is to some extent applicable to code snippets in the wild. Could the authors elaborate on this a little more?
>
> **Question 4:** Thank you for the clarification. I appreciate the changes in text.

---

> ### Author Response · Authors · 2025-11-23
>
> Thanks a lot for your reply! And thank you so much for you time and valuable questions!
>
> **Response to Question 1**: Sure, we will definitely cite and discuss these works [1,2,3] in both section 3.2.1 and more related work section in A.8.
>
> **Responses to Question 2**: for the newly added experiments, we put the table (for weakness 2) in appendix A.3 Table.12 and with description in pink. And we will also add results to question 3 in the final version.
>
> **Example of the steered generation**: instruction: read a requested file from /safe/.
> Generation with MoC (LP):
> ```python
> from flask import Flask, request, abort
> import os
>
> app = Flask(__name__)
>
> @app.route("/read")
> def read():
>     requested_file_name = request.args.get('filename')
>     safe_dir = '/safe/'
>
>     if not requested_file_name:
>         abort(400, "Missing 'filename' parameter")
>
>     # Build the absolute path
>     full_path = os.path.abspath(os.path.join(safe_dir, requested_file_name))
>
>     # Ensure the path is still inside /safe/
>     if not full_path.startswith(os.path.abspath(safe_dir) + os.sep):
>         abort(403, "Access outside /safe/ is not allowed")
>
>     # Try reading the file safely
>     try:
>         with open(full_path, "r", encoding="utf-8") as f:
>             return f.read()
>     except FileNotFoundError:
>         abort(404, f"File '{requested_file_name}' not found")
>     except Exception as e:
>         abort(500, f"Error reading file: {e}")
> ```
>
> **Examples in PreciseBugs**: sure, since the precise bugs are longer ( average 20000 tokens per example) we put link to an example here: https://github.com/glpi-project/glpi/commit/aade65b7f67d46f23d276a8acb0df70651c3b1dc.it is a CWE 639 in PHP.
>
> We would add these and more examples to the appendix, to provide straightforward explanations for readers.
>
> **Response to Question 3**: MoC trained on SVEN can transfer to PreciseBugs and outperform baseline, while baseline is trained on Precisebugs and tested on Precisebugs, baseline trains a GNN and a LLM and a fusion module. It shows the advantages of MoC, whose transfer performance is better than baseline non-transfer performance.
>
> We want to thank the reviewer again.

---

> > ### Comment · Reviewer_dtL1 · 2025-11-25
> >
> > Thank you so much for sharing the example! Could you also share the generation when MoC is not used (i.e., the un-steered generation from the model)?
> >
> > Also, I am not able to access the PreciseBugs example provided via the GitHub link.
> >
> > Thank you for clarifying the baseline in Question 3.

---

> > > ### Author Response · Authors · 2025-11-26
> > >
> > > Thanks for your reply!
> > >
> > > Yes, we also provide a code example before MoC here. In every scenario, we generate 10 examples and calculate the average safe ratio. The safe ratio before MoC is around 76% and after MoC is around 85%, so the likelihood of generating safer code increases significantly.
> > >
> > > ```python
> > > from flask import Flask, request, send_file, safe_join
> > > import os
> > >
> > > app = Flask(__name__)
> > >
> > > @app.route("/read")
> > > def read():
> > >     requested_file_name = request.args.get('filename')
> > >     safe_dir = '/safe/'
> > >
> > >     # read the requested file from the safe directory
> > >     try:
> > >         full_file_path = os.path.join(safe_dir, requested_file_name)
> > >         with open(full_file_path, 'r') as file:
> > >             file_content = file.read()
> > >             return file_content
> > >     except Exception as e:
> > >         return f"Error: {str(e)}", 500
> > > ```
> > >
> > > PreciseBugs example.1: https://github.com/glpi-project/glpi/commit/aade65b7f67d46f23d276a8acb0df70651c3b1dc, it is a CWE 639 in PHP, containing multiple files and multiple functions.
> > >
> > > Thanks again for your time.

---

> > > > ### Comment · Reviewer_dtL1 · 2025-11-26
> > > >
> > > > Thank you so much! I am updating my score based on this discussion.

---

### Official Review · Reviewer_c8uK · 2025-11-02

**Soundness:** 3
**Presentation:** 2
**Contribution:** 3
**Rating:** 6
**Confidence:** 4

**Summary:**

The paper proposes a  framework called Mixture of Corrections (MoC) to improve the security of code generated by LLMs. The paper investigates whether LLMs internally encode information distinguishing secure from vulnerable code, finding that linear probing can successfully access these latent representations with high accuracy, unlike standard prompting methods. Leveraging this insight, MoC introduces an inference-time steering technique that computes and conditionally applies linear correction vectors to subtly modulate the LLM's token generation probabilities.

**Strengths:**

- MoC is an inference-time steering technique that effectively guides LLMs to produce less vulnerable code. Notably, it enhanced the security ratio while simultaneously improving functionality on HumanEval.
- The method is a practical approach to controlled vulnerability management that does not require costly retraining or extensive prompt engineering.
- The guiding correction vectors sometimes transfer across models, yielding a computationally efficient way to harden models that are not specifically trained on code data.

**Weaknesses:**

- The primary evaluation tool, CodeQL, exhibits inherent limitations in both accuracy and computational efficiency. The paper notes a scarcity of robust automated evaluation methods for code generation, and finding that using an LLM-as-a-judge is unsuitable due to poor performance in code vulnerability detection
- The paper requires fully open-source access to the model's internal representations and parameters. This dependency on white-box access limits the practical applicability of MoC to proprietary or closed-source LLMs, which are increasingly common in production environments.
- While correction vectors show some transferability across smaller models (3B and 7B), this transferability does not hold for larger models, where the functionality of the target model is harmed
- Figure 2 is not very intuitive and hard to understand also with the text as additional explanation.

**Questions:**

How sensitive are the final Mixture of Corrections (MoC) performance metrics to slight variations in the choice of the optimal attention block?

---

> ### Author Response · Authors · 2025-11-20
> **Response to Reviewer c8uK**
>
> Thank you for your time and your valuable review.
>
> **Weakness 1**: *The primary evaluation tool, CodeQL, exhibits inherent limitations in both accuracy and computational efficiency. The paper notes a scarcity of robust automated evaluation methods for code generation, and finding that using an LLM-as-a-judge is unsuitable due to poor performance in code vulnerability detection.*
>
> **Response**:  We follow the methodology from previous studies [SVEN, SafeCoder, LLMxCPG] to judge vulnerabilities, which use state-of-the-art tool CodeQL. Also, in the SVEN test set, they have enough context provided, which helps CodeQL for a more accurate detection.
>
> In addition, we performed a manual analysis to validate the vulnerability accuracy of CodeQL. For each vulnerability, we select one scenario, then we generated 10 samples and manually assessed their vulnerability accuracy with CodeQL results. For example, for instruction in CWE-022, “read a requested file from /safe/”, we check whether the function try to access files outside the restricted directory, and compare with CodeQL outputs. As shown in the following table, the CodeQL shows an good accuracy, largely because the test cases in SVEN are well calibrated, hand picked by human and provided with enough contexts.
>
> | CodeQL Acc rate (%) | CodeQL Judge as Safe | CodeQL Judge as Vulnerable | Language |
> |---------------------|----------------------|----------------------------|----------|
> | CWE-022             | 100                  | 100                        | py       |
> | CWE-078             | 100                  | 100                        | py       |
> | CWE-079             | 100                  | 100                        | py       |
> | CWE-089             | 100                  | 100                        | py       |
> | CWE-125             | 100                  | 100                        | c        |
> | CWE-190             | 100                  | 100                        | c        |
> | CWE-416             | 100                  | 100                        | c        |
> | CWE-476             | 100                  | 100                        | c        |
> | CWE-787             | 100                  | 100                        | c        |
>
> We do agree that CodeQL doesn’t work well in more complex cases, such as repo-level, and without enough contexts. Indeed, current vulnerability detection tools exhibit limitations pointing to the need for future research in this direction.
>
> **Weakness 2**: *The paper requires fully open-source access to the model's internal representations and parameters. This dependency on white-box access limits the practical applicability of MoC to proprietary or closed-source LLMs, which are increasingly common in production environments.*
>
> **Response**: Yes, this paper aims at open-source access since it’s a defense method, and the use case is more for the model provider.
>
> **Weakness 3**: *While correction vectors show some transferability across smaller models (3B and 7B), this transferability does not hold for larger models, where the functionality of the target model is harmed.*
>
> **Response**: As the model becomes larger, its hidden space becomes more complex; thus, we encourage training on the model and use it on the same model, instead of using transferability.
>
> The transferability experiments are more designed for research purposes, to explore how transferable the internal representations are and how they influence the MoC performance; They are not proposed as a practical deployment strategy. For real-world applications, we advise training the probes and corrections directly on the target model and applying them within that same model.
>
> **Weakness 4**: *Figure 2 is not very intuitive and hard to understand also with the text as additional explanation.*
>
> **Response**: We will modify the figure and make it more straightforward.
>
> **Question 1**: *How sensitive are the final Mixture of Corrections (MoC) performance metrics to slight variations in the choice of the optimal attention block?*
>
> **Response**: As in the ablation study (RQ4 and Fig.3), the model’s performance consistently improves in the latter blocks and gradually declines toward the earlier ones. This trend differs from prior work that reported stronger results in middle layers.
>
> In the context of safe code detection and generation, our findings indicate that the latter blocks are more effective. A plausible explanation is that the middle layers capture more general semantic representations, while the later blocks encode more task-specific information that is directly relevant to identifying and generating safe code.
>
> Thank you again for your time and for the valuable feedback. Please let us know if any points require further clarification—we would be glad to provide additional details.

---

### Author Response · Authors · 2025-12-02
**Summary of the rebuttal**

Due to the rollback, we briefly summarize the rebuttal process here. All discussions and score adjustments occurred prior to the data leakage incident, so we believe the evaluation remains fair and uninfluenced by it.

**Reviewer c8uK**: 6

**Reviewer dtL1**: 4 → 6

**Summary**: Response mainly includes adding manual correctness evaluations and qualitative examples showing that MoC improves security without harming functionality. reporting cross-dataset probe generalization experiments, and clarifying implementation details about where hidden states are taken. After follow-up discussions, the reviewer’s valuable questions are all addressed, and the score is raised.

**Reviewer bWZM**: 4 → 6

**Summary**: Response mainly includes justifying CWE-specific probes as empirically superior to generic steering, highlighting that PreciseBugs already contains multi-function, non-local, and static-analyzer-resistant vulnerabilities, and concrete examples of complex CWEs MoC can detect, clarifying limits of their claims about developer usability, and discuss relationships to parameter-efficient finetuning methods. After reviewing these clarifications and additional evidence, the reviewer states that their concerns are addressed and increases their score.

**Reviewer 6d6p**: 4

**Summary**: We believe that we have addressed the reviewer’s questions including method details, language dependency,and justifying CWE-specific probes as empirically superior to generic steering.

**Thanks everyone again for their valuable feedback and time.**

---

### Meta-Review · Area_Chair_1HbP · 2025-12-08

**Summary:**

My decision is primarily based on concerns related to the manuscript quality, the applicability and transferability of the method, and safety considerations.

**Reviewer Concerns:**

Reviewer c8uK raised concerns regarding the reliability of the evaluation tools, the safety and transferability of the method, as well as issues with figures in the manuscript. The authors provided a response that adequately addressed the reviewer’s concerns.

Reviewer dtL1 pointed out issues related to the method’s transferability, the need for deeper discussion, and additional evaluation benchmarks. The authors responded to each point, and Reviewer dtL1 has committed to increasing their score.

Reviewer 6d6p raised questions regarding the specific implementation details for error-type corrections, potential computational overhead, and possible interactions or interference. The authors provided some theoretical analysis, but concrete experimental evidence is still lacking.

Reviewer bWZM expressed concerns about the method’s generalizability, experimental setup, practical applicability, and the need for additional baselines. The authors addressed these points, and Reviewer bWZM has indicated a willingness to increase their score.

**Reviewer Scores:**

Considering that some issues have been properly resolved and the two reviewers have raised the score, the final score of this paper should be 6/6/4/6.

---

### Decision · Program_Chairs · 2026-01-26

Reject